# Pharmacokinetic/Pharmacodynamic Target Attainment of Continuous Infusion Piperacillin–Tazobactam or Meropenem and Microbiological Outcome among Urologic Patients with Documented Gram-Negative Infections

**DOI:** 10.3390/antibiotics12091388

**Published:** 2023-08-31

**Authors:** Pasquale Maria Berrino, Milo Gatti, Matteo Rinaldi, Eugenio Brunocilla, Pierluigi Viale, Federico Pea

**Affiliations:** 1Division of Urology, IRCCS Azienda Ospedaliero-Universitaria of Bologna, 40138 Bologna, Italy; pasquale.berrino@studio.unibo.it (P.M.B.); eugenio.brunocilla@unibo.it (E.B.); 2Department of Medical and Surgical Sciences, Alma Mater Studiorum University of Bologna, 40138 Bologna, Italy; mat.rinaldi1989@gmail.com (M.R.); pierluigi.viale@unibo.it (P.V.); federico.pea@unibo.it (F.P.); 3Clinical Pharmacology Unit, Department for integrated Infectious Risk Management, IRCCS Azienda Ospedaliero-Universitaria of Bologna, 40138 Bologna, Italy; 4Infectious Disease Unit, Department for integrated Infectious Risk Management, IRCCS Azienda Ospedaliero-Universitaria of Bologna, 40138 Bologna, Italy

**Keywords:** piperacillin–tazobactam, meropenem, urology, Gram-negative infections, PK/PD target attainment, microbiological outcome

## Abstract

(1) Objectives: To describe the relationship between pharmacokinetic/pharmacodynamic (PK/PD) target attainment of continuous infusion (CI) piperacillin–tazobactam or meropenem monotherapy and microbiological outcome in a case series of urological patients with documented Gram-negative infections. (2) Methods: Patients admitted to the urology ward who were treated with CI piperacillin–tazobactam or meropenem monotherapy for documented Gram-negative infections and underwent real-time therapeutic drug monitoring (TDM)-guided expert clinical pharmacological advice (ECPA) program from June 2021 to May 2023 were retrospectively retrieved. Average steady-state (C_ss_) piperacillin–tazobactam and meropenem concentrations were determined, and the free fractions (*f*C_ss_) were calculated. Optimal PK/PD target attainments were defined as an *f*C_ss_/MIC ratio >4 for CI meropenem and an *f*C_ss_/MIC ratio of piperacillin >4 coupled with an *f*C_ss_/C_T_ ratio for tazobactam >1 for piperacillin–tazobactam (joint PK/PD target). The relationship between beta-lactam PK/PD targets and microbiological outcome was explored. (3) Results: Sixteen urologic patients with documented Gram-negative infections (62.5% complicated urinary tract infections (cUTI)) had 30 TDM-guided ECPAs. At first TDM assessment, beta-lactam dosing adjustments were recommended in 11 out of 16 cases (68.75%, of which 62.5% decreases and 6.25% increases). Overall, beta-lactam dosing adjustments were recommended in 14 out of 30 ECPAs (46.6%). Beta-lactam PK/PD target attainments were optimal in 100.0% of cases. Microbiological failure occurred in two patients, both developing beta-lactam resistance. (4) Conclusion: A TDM-guided ECPA program may allow for optimizing beta-lactam treatment in urologic patients with documented Gram-negative infections, ensuring microbiological eradication in most cases.

## 1. Introduction

Urologic patients may have an increased risk of developing healthcare-associated infections [1]. In this scenario, surgical procedures performed by means of endourological transurethral access, wide use of double-J stent, and high prevalence of urinary catheter carriers both prior to and during hospital admission may represent major risk factors for developing healthcare-associated urinary tract infections (UTIs) [1,2,3]. Among hospitalized urologic patients, complicated UTIs (cUTIs) may account for up to 60–70% of infections [1,2], and may benefit from tailored antimicrobial therapy.

In recent years, multidrug-resistant (MDR) Gram-negative pathogens causing cUTIs have started rapidly emerging and spreading globally [1,2,4,5,6]. Consequently, an in depth knowledge of the local epidemiology in the urologic setting is essential for determining empirical antimicrobial therapy of Gram-negative cUTIs [2]. Piperacillin–tazobactam and meropenem are currently the most recommended agents in cUTIs according to their good activity against extended-spectrum beta-lactamase (EBSL)-producing *Enterobacterales* [2].

Nowadays, adequate source control and pharmacokinetic/pharmacodynamic (PK/PD) optimization of antibiotic therapy are considered two mainstays in the management of Gram-negative infections [7]. Aggressive PK/PD targets of beta-lactams, namely 100% of the dosing interval with free beta-lactam concentration at least 4-fold above the minimum inhibitory concentration (MIC) of the targeted pathogen, were shown to both maximize clinical efficacy and to suppress resistance emergence [8,9,10,11]. Therapeutic drug monitoring (TDM) may be a helpful tool for granting optimal PK/PD target attainment during beta-lactam therapy [12]. Notably, several studies and systematic reviews recently supported the role of a TDM-guided approach for optimizing beta-lactam therapy compared to standard management in significantly improving clinical and microbiological outcomes in different clinical scenarios [13,14,15,16,17]. 

The aim of this study was to investigate the role of a TDM-guided approach in optimizing PK/PD target attainment of CI piperacillin–tazobactam or meropenem in a case series of hospitalized urologic patients affected by documented Gram-negative infections and to describe the relationship with microbiological outcome.

## 2. Results

Overall, 16 urologic patients with documented Gram-negative infections receiving TDM-guided CI piperacillin–tazobactam (n = 10) or meropenem (n = 6) were included in the study. Demographics and clinical features of the patients are reported in Table 1. Case-by-case assessment of patients receiving piperacillin–tazobactam and meropenem are shown in Table 2 and Table 3, respectively. 

The median (interquartile range (IQR)) age was 70.5 years (65.25–74.0 years), with a male preponderance (68.75%). The median (IQR) Charlson Comorbidity Index (CCI) was 5 (3.75–8). The median (IQR) average baseline creatinine clearance (CLcr) was 41 mL/min/1.73 m^2^ (28.0–74.25 mL/min/1.73 m^2^), and 10 out of the 16 included patients (62.5%) showed an average baseline CLcr below 50 mL/min/1.73 m^2^. None of the included patients developed sepsis-related acute kidney injury. Bladder cancer (43.75%) and hydronephrosis (31.25%) were the main causes for hospital admission. Surgical intervention was required in 11 out of the 16 cases (68.75%).

The types of infection were cUTIs in 6/16 cases, cUTI + bloodstream infection (BSI) and BSIs in 4/16 cases each, and surgical site infection (SSI) and necrotizing soft tissue infection (NSTI) in 1/16 cases each. Overall, 19 Gram-negative pathogens were isolated, *Escherichia coli* (36.7%), *Pseudomonas aeruginosa* (21.1%), and *Klebsiella pneumoniae* (15.8%) being the most frequent. ESBL- and/or AmpC-producing isolates accounted for 7 out of 19 (36.8%) isolates.

CI piperacillin–tazobactam was used in 10/16 cases (62.5%), and meropenem in the other six patients (37.5%). Median (IQR) daily maintenance dose (MD) was 13,500 mg (9000–13,500 mg) and 2000 mg (2000–2000 mg) for piperacillin–tazobactam and meropenem, respectively. Median (IQR) free steady-state concentrations (*f*C_ss_) were 11.4 mg/L (7.0–23.0 mg/L), 55.8 mg/L (35.4–80.4 mg/L), and 7.4 mg/L (5.2–12.7 mg/L) for meropenem, piperacillin, and tazobactam, respectively.

A total of 30 TDM-guided ECPAs were performed, with a median (IQR) of 2 (1–2) per patient. At first TDM assessment, beta-lactam dosing adjustments were recommended in 11 out of 16 cases (68.75%, of which 62.5% were decreases and 6.25% were increases). Overall, beta-lactam dosing adjustments were recommended in 14 out of 30 ECPAs (46.6%, of which 3.3% were increases and 43.3% were decreases). The average *f*C_ss_/MIC ratios were optimal in all six cases (100.0%) treated with meropenem (Figure 1). Similarly, all patients treated with piperacillin–tazobactam (100.0%) attained optimal joint PK/PD targets (Figure 1).

Microbiological eradication was achieved in 14 out of 16 cases (87.5%). Microbiological failure with resistance development occurred in two patients (12.5%), namely, one case of bacteraemic cUTI caused by *Klebsiella pneumoniae* treated with piperacillin–tazobactam (relapse due to an ESBL-producing *Klebsiella pneumoniae* isolate) and one case of cUTI due to AmpC-producing *Serratia marcescens* treated with meropenem (relapse due to a KPC-producing *Serratia marcescens* strain). Clinical cure was achieved in 13 out of 16 cases (81.25%). None of the included patients died at 30-day follow-up.

## 3. Discussion

To the best of our knowledge, this is the first study that described the relationship of PK/PD target attainment of CI piperacillin–tazobactam or meropenem and microbiological outcome in a case-series of hospitalized urologic patients treated for documented Gram-negative infections. 

Our findings showed that TDM-guided CI of piperacillin–tazobactam and meropenem allowed in all of the included patients optimal PK/PD target attainment against ESBL- and/or AmpC-producing *Enterobacterales* even with borderline susceptibility, possibly contributing to maximization of microbiological eradication in the vast majority of cases. The large proportion of cUTI (exceeding 60% of cases) in our cohort was consistent with previous studies conducted in the urologic setting and the non-negligible proportion of ESBL- and/or AmpC-producing *Enterobacterales* as causative pathogens was related to the fact that most patients underwent surgical procedures (approximatively 70%) mainly due to bladder cancer [2].

The study may support the relevance that a real-time TDM-guided ECPA approach may have even in the surgical setting, as recently suggested [18]. CI is the best administration mode for attaining aggressive PK/PD targets with beta-lactam under the same daily dose [19,20,21,22,23]. Furthermore, some studies reported significantly lower mortality rates, higher clinical cure rates, and higher microbiological eradication rates with the administration of beta-lactams via CI compared to intermittent infusion [19,20,21,22]. Noteworthy a large proportion of our urologic patients were elderly with moderate-to-severe renal dysfunction, and the TDM-guided approach of CI piperacillin–tazobactam and meropenem allowed optimal PK/PD target attainment, even by consistently decreasing dosages in several cases, thus minimizing clinically unnecessary drug overexposure and the risk of concentration-dependent neurotoxicity [24,25]. 

In regard to piperacillin–tazobactam, we first introduced the concept of joint PK/PD target attainment based on measuring both piperacillin and tazobactam concentrations. Previous real-time studies tailored piperacillin–tazobactam therapy based only on measuring piperacillin concentrations [16,17]. Indeed, this approach could prove to be inadequate considering that in a hollow-fiber infection model, escalating tazobactam concentrations with fixed piperacillin exposure resulted in decreased piperacillin MIC against ESBL-producing *Escherichia coli* and *Klebsiella pneumoniae* isolates [26]. This may support the contention that the PK/PD target of piperacillin–tazobactam against ESBL-producers should be based not only on piperacillin but also on tazobactam concentrations, as just previously suggested for other beta-lactam/beta-lactamase combinations, that is, ceftazidime/avibactam [27,28]. Furthermore, it is noteworthy that CI administration may be a useful tool for maintaining tazobactam concentrations steadily over time above the safeguarded fixed threshold of 4 mg/L, as reported in our cases.

The limitations of our study have to be addressed. The retrospective monocentric study design and the limited sample size must be acknowledged. Total meropenem and piperacillin–tazobactam concentrations were determined, and free fractions were estimated based on the plasma protein binding retrieved in the literature. However, the potential impact of hypoalbuminemia in affecting free antibiotic concentrations may be negligible considering that both agents have low-to-moderate plasma protein binding. A point of strength of our study is the fact that this is the first real-life experience describing the relationship between PK/PD target attainment of CI piperacillin–tazobactam and/or meropenem and microbiological outcome in the challenging scenario of hospitalized urologic patients with documented Gram-negative infections.

## 4. Materials and Methods

### 4.1. Study Design

We performed a retrospective case series of hospitalized patients admitted and managed during hospitalization in the urology ward of the IRCCS Azienda Ospedaliero-Universitaria of Bologna, Italy, in the period between 1 June 2021 and 31 May 2023 who were treated with CI piperacillin–tazobactam or meropenem monotherapy for documented Gram-negative infections and underwent real-time therapeutic drug monitoring (TDM). 

The study was conducted according to the Declaration of Helsinki and was approved by the Ethical Committee of the IRCCS Azienda Ospedaliero-Universitaria of Bologna (no. 442/2021/Oss/AOUBo, approved on 28 June 2021). Signed informed consent was waived due to the retrospective and observational nature of the investigation, according to hospital agreements.

### 4.2. Data Collection

For each case, demographic (age, sex weight, height, body mass index (BMI)) and clinical/laboratory data (admission diagnosis, comorbidity, need for surgical intervention, CLCr at baseline and during treatment) were retrieved. CCI was calculated for each patient. Bacterial clinical isolates with MIC values of piperacillin–tazobactam or meropenem, type/site of infection, piperacillin–tazobactam or meropenem dosage, average piperacillin–tazobactam or meropenem concentrations, treatment duration, overall number of ECPAs, ECPA-recommended dosing adjustments, ECPA-recommended dosing adjustments at first TDM assessment, and microbiological outcome were collected. 

Sites of infections were defined according to Centers for Disease Control and Prevention (CDC) criteria [29]. Documented BSI was defined as the isolation of a Gram-negative pathogen from at least one blood culture [29]. cUTI was defined as the presence of local and systemic signs and/or symptoms coupled with the isolation of a Gram-negative pathogen (≥10^5^ microorganisms per cc of urine) from urine culture with no more than two different species of microorganisms isolated from the same sample [29]. SSI was defined as the isolation of a Gram-negative pathogen collected near or at the incision site and/or deeper underlying tissue spaces and organs within 30 days of a surgical procedure [30]. NSTI was defined as the isolation of Gram-negative pathogens from a biopsied sample of the advancing margin skin lesion involving the superficial fascia and subfascial tissue [29].

### 4.3. Beta-Lactam Administration and Sampling Procedure

Piperacillin–tazobactam and/or meropenem were prescribed at the discretion of the treating physician and/or the infectious disease consultant in terms of therapeutic indication, starting dosage, and treatment duration. Specifically, piperacillin–tazobactam was always started as first-line therapy in patients with suspected or documented Gram-negative infection, excepted in cases with documented severe and/or life-threatening allergy to penicillin or with recent colonization due to ESBL-producing *Enterobacterales*. Targeted therapy with piperacillin–tazobactam was maintained if feasible according to the results of the susceptibility tests, or otherwise shifted to meropenem.

Piperacillin–tazobactam was started with a loading dose (LD) of 9 g over 2 h infusion, whereas meropenem was started with an LD of 2 g over 2 h infusion, followed by an initial MD administered by CI. The initial MD regimen was defined according to renal function and underlying pathophysiological conditions of each included patient, and subsequently optimized by means of a real-time TDM-guided ECPA approach. Specifically, the initial piperacillin–tazobactam MD was of 16/2 g/day, 12/1.5 g/day, or 8/1 g/day in patients with a CLcr > 40 mL/min/1.73 m^2^, 20–40 mL/min/1.73 m^2^, or < 20 mL/min/1.73 m^2^, respectively. In regard to meropenem, 500–1000 mg q6h over 6 h infusion represented the initial MD in all of the classes of renal function, except for patients with severe renal dysfunction (i.e., CLcr < 20 mL/min/1.73 m^2^), in whom the initial MD was 250 mg q6h over 6 h infusion.

For granting properly CI, aqueous solutions and meropenem were reconstituted and infused every 24 h over 24 h for piperacillin–tazobactam and every 6–8 h over 6–8 h for meropenem, as recommended [31,32,33]. 

Blood samples for measuring piperacillin–tazobactam or meropenem C_ss_ were collected firstly after at least 24 h from starting therapy and then reassessed whenever feasible. Total piperacillin–tazobactam and meropenem plasma concentrations were determined by means of a validated liquid chromatography–tandem mass spectrometry method [10]. 

TDM results of piperacillin–tazobactam or meropenem underwent real-time expert interpretation by the MD clinical pharmacologists (ECPA) who suggested dosing adaptation whenever needed. The TDM-guided ECPA was structured as previously reported [18,34].

### 4.4. Relationship between Beta-Lactam PK/PD Target and Microbiological Outcome

In regard to meropenem, percentage of time above the MIC was selected as PK/PD parameter of efficacy and defined as *f*C_ss_/MIC. Total meropenem concentrations were measured and, according to a plasma protein binding of 2% reported in the literature [35], the *f* was calculated multiplying total meropenem C_ss_ by 0.98. PK/PD target attainment, which was defined as optimal when *f*C_ss_/MIC ratio was >4 and quasi-optimal or suboptimal when *f*C_ss_/MIC ratio was 1–4 or <1, respectively, as previously reported [36]. In patients performing more than one TDM-guided ECPA for meropenem dosing personalization, the average *f*C_ss_ was calculated as the mean of all the meropenem C_ss_ values assessed (the first one before any dosage adjustment and the subsequent ones after eventual dosage adjustments).

In regard to piperacillin–tazobactam, a joint PK/PD target was defined. Percentage of time above the MIC was selected as the PK/PD parameter of efficacy of piperacillin (*f*C_ss_/MIC ratio), whereas the ratio between tazobactam C_ss_ and the threshold of 4 mg/L (C_T_) (*f*C_ss_/C_T_ ratio) was selected as the PK/PD parameter of efficacy of tazobactam. The threshold concentration of 4 mg/L was selected according to the value adopted by the EUCAST for tazobactam when testing piperacillin–tazobactam susceptibility. Total piperacillin and tazobactam concentrations were measured, and the *f* values were calculated by multiplying total piperacillin and tazobactam C_ss_ by 0.80 and 0.77, respectively, according to a plasma protein binding of 20% and 23% [37]. The joint PK/PD target attainment of piperacillin–tazobactam was defined as optimal if both the piperacillin *f*C_ss_/MIC ratio was >4 and the tazobactam *f*Css/C_T_ ratio was >1 and quasi-optimal or suboptimal if only one or none of the two thresholds were attained, respectively, likewise reported for other beta-lactam/beta-lactamase inhibitor combinations [38]. In patients performing more than one TDM-guided ECPA for piperacillin–tazobactam dosing personalization, average *f*C_ss_ was calculated as the mean of all the piperacillin and tazobactam C_ss_ values assessed (the first one before any dosage adjustment and the subsequent ones after eventual dosage adjustments).

These thresholds were based on evidence reported in in vitro studies, experimental animal models, and clinical studies showing that the attainment of aggressive PK/PD targets consisting of beta-lactam C_ss/_MIC ratios ≥ 4 (equivalent to 100% *f*T_> 4×MIC_) may be associated with increased microbiologic eradication and suppression of resistance development [8,9,10,11]. 

The MICs of the beta-lactams against isolated Gram-negative pathogens (*Enterobacterales* and/or *Pseudomonas aeruginosa*) were measured by means of a semi-automated broth microdilution method (Microscan Beckman NMDRM1) and interpreted according to the European Committee on Antimicrobial Susceptibility Testing (EUCAST) clinical breakpoints [39]. Resistance was defined as the isolation of a Enterobacterales or P. aeruginosa with an MIC > 8 mg/L or 2 mg/L for piperacillin–tazobactam or meropenem, respectively.

The relationship between beta-lactam PK/PD target attainment and microbiological outcome was assessed in each patient. Microbiological eradication was defined as the absence of the Gram-negative pathogen isolated in the index cultures in at least one subsequent culture after starting piperacillin–tazobactam or meropenem therapy. Resistance development was defined as an increase in the piperacillin–tazobactam or meropenem MIC against the original clinical isolate beyond the EUCAST clinical breakpoint of susceptibility. Clinical cure was defined as the complete resolution of signs and symptoms of the infection coupled with documented microbiological eradication at the end of treatment and absence of recurrence or relapse at 30-day follow up, piperacillin–tazobactam or meropenem resistance development, and 30-day mortality rate.

### 4.5. Statistical Analysis

Descriptive statistics were used to describe the patient sample. Median and interquartile range (IQR) was used for expressing continuous data, whereas counts or percentages were used for presenting categorical variables. Statistical analysis was performed by using MedCalc for Windows (MedCalc statistical software, version 19.6.1, MedCalc Software Ltd., Ostend, Belgium).

## 5. Conclusions

In conclusion, our findings indicated that TDM-guided CI piperacillin–tazobactam and meropenem may be helpful in attaining optimal joint PK/PD targets among urologic elderly patients with documented Gram-negative infections and renal dysfunction. This approach could contribute to microbiological eradication in such cases. Large prospective clinical studies are warranted for confirming our findings in this clinical scenario.

## Figures and Tables

**Figure 1 antibiotics-12-01388-f001:**
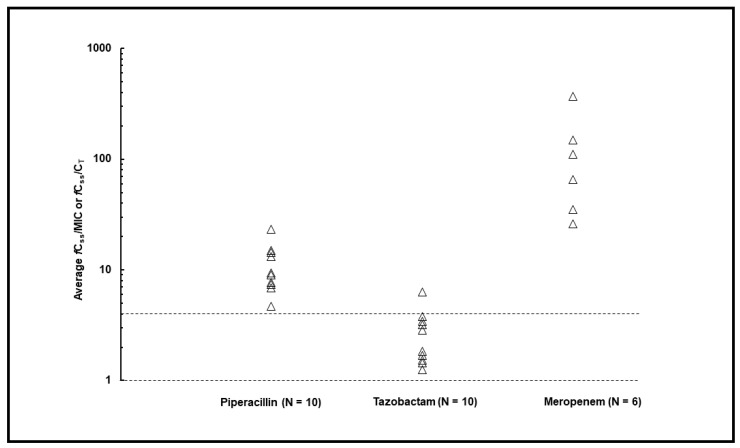
PK/PD target attainment for piperacillin–tazobactam and meropenem in included patients (N = 16). Dotted lines represent optimal *f*C_ss_/MIC > 4 for piperacillin and meropenem, and *f*C_ss_/C_T_ > 1 for tazobactam. White triangles: attainment of optimal PK/PD targets.

**Table 1 antibiotics-12-01388-t001:** Demographics and clinical characteristics of included patients.

Demographics and Clinical Variables	Patients (N = 16)
Patient demographics	
Age (years) (median (IQR))	70.5 (65.25–74.0)
Gender (male/female) (n (%))	11/5 (68.75/31.25)
Body weight (Kg) (median (IQR))	71.5 (64.25–81.25)
Body mass index (Kg/m^2^) (median (IQR))	25.2 (23.7–27.7)
Charlson Comorbidity Index (median (IQR))	5 (3.75–8)
Average baseline creatinine clearance (mL/min/1.73 m^2^) (median (IQR))	41.0 (28.0–74.25)
Average baseline serum albumin levels (mg/dL) (median (IQR))	3.06 (2.99–3.30)
Average baseline white blood cell count (×10^3^ µL) (median (IQR))	11.51 (8.48–23.00)
Average baseline serum CRP levels (mg/dL) (median (IQR))	21.65 (14.21–26.21)
Average baseline serum PCT levels (ng/mL) (median (IQR))	7.00 (1.08–64.43)
Surgical intervention (n (%))	11 (68.75)
Site of infection (n (%))	
cUTI	6 (37.5)
cUTI + BSI	4 (25.0)
BSI	4 (25.0)
SSI	1 (6.25)
NSTI	1 (6.25)
Isolated gram-negative pathogens ^1^ (n (%))	
*Escherichia coli*	7 (36.7)
*Pseudomonas aeruginosa*	4 (21.1)
*Klebsiella pneumoniae*	3 (15.8)
*Enterobacter cloacae*	2 (10.5)
*Klebsiella oxytoca*	1 (5.3)
*Serratia marcescens*	1 (5.3)
*Bacteroides fragilis*	1 (5.3)
*Beta-lactam treatment*	
Meropenem (n (%))	6 (37.5)
Piperacillin–tazobactam (n (%))	10 (62.5)
Beta-lactam TDM	
Meropenem dose (mg/day) (median (IQR))	2000 (2000–2000)
Piperacillin–tazobactam dose (mg/day) (median (IQR))	13,500 (9000–13,500)
Meropenem *f*C_ss_ (mg/L) (median (IQR))	11.4 (7.0–23.0)
Piperacillin *f*C_ss_ (mg/L) (median (IQR))	55.8 (35.4–80.4)
Tazobactam *f*C_ss_ (mg/L) (median (IQR))	7.4 (5.2–12.7)
Meropenem PK/PD target attainment	
*f*C_ss_/MIC > 4 (n (%))	6 (100.0)
*f*C_ss_/MIC = 1–4 (n (%))	0 (0.0)
*f*C_ss_ or C_min_/MIC < 1 (n (%))	0 (0.0)
Piperacillin–tazobactam joint PK/PD target attainment	
*f*C_ss_/MIC > 4 (n (%))	10 (100.0)
*f*C_ss_/MIC = 1–4 (n (%))	0 (0.0)
*f*C_ss_ or C_min_/MIC < 1 (n (%))	0 (0.0)
Expert clinical pharmacological advice	
Overall ECPAs	30
No. of TDM-guided ECPA per patient (median (IQR))	2 (1–2)
No. of dosage confirmed (n (%))	16 (53.4)
No. of dosage increases (n (%))	1 (3.3)
No. of dosage decreases (n (%))	13 (43.3)
First TDM assessment within desired range (n (%))	5 (31.25)
First TDM increase (n (%))	1 (6.25)
First TDM decrease (n (%))	10 (62.5)
Microbiological outcome	
Microbiological eradication (n (%))	14 (87.5)
Microbiological failure (n (%))	2 (12.5)
Resistance development (n (%))	2 (12.5)
Clinical cure (n (%))	13 (81.25)

^1^ A total of 19 pathogens were isolated. BSI: bloodstream infection; C_min_: trough concentration; CRP: C-reactive protein; C_ss_: steady-state concentration; C_T_: threshold concentration; cUTI: complicated urinary tract infection; ECPA: expert clinical pharmacological advice; IQR: interquartile range; MIC: minimum inhibitory concentration; PCT: procalcitonin; SSI: surgical site infection; TDM: therapeutic drug monitoring.

**Table 2 antibiotics-12-01388-t002:** Case-by-case demographic and clinical features of 10 urologic patients with documented Gram-negative infections treated with CI piperacillin–tazobactam according to a TDM-guided ECPA program.

ID Case	Age/Sex	CCI	Hospital Admission Diagnosis	Average CLCr (mL/min/ 1.73 m^2^)	Type of Infection	Pathogen	MIC (mg/L)	Beta-Lactam Initial Dosing	*f*Css/MIC Ratio	*f*Css/C_T_ Ratio	Joint PK/PD Target	ECPA Recommendation at First TDM	Microbiological Eradication	Clinical Outcome
#1	82/M	8	Bladder cancer	30.3	cUTI	*P. aeruginosa*	8	PIP-TZB 13.5 g/day CI	7.35	3.18	Optimal	Decrease	Yes	Cured
#2	67/M	2	Inflammation of renal cyst	69.7	cUTI	*E. coli*	8	PIP-TZB 18 g/day CI	7.68	1.45	Optimal	Confirm	Yes	Cured
#3	48/F	3	Hydronephrosis	18	cUTI + BSI	*E. coli*	4	PIP-TZB 13.5 g/day CI	15.07	2.85	Optimal	Decrease	Yes	Cured
#4	68/M	8	Macrohematuria	42	BSI	ESBL-producing *E. coli*	8	PIP-TZB 13.5 g/day CI	9.41	3.79	Optimal	Decrease	Yes	Cured
#5	73/M	5	Hydronephrosis	37	cUTI	*E. coli*	4	PIP-TZB 13.5 g/day CI	9.02	1.25	Optimal	Confirm	Yes	Cured
#6	74/M	5	Hydronephrosis	20	cUTI + BSI	*P. aeruginosa*	8	PIP-TZB 13.5 g/day CI	14.30	3.43	Optimal	Decrease	Yes	Failed
#7	48/M	3	Renal transplant	31	cUTI + BSI	*K. pneumoniae*	4	PIP-TZB 9 g/day CI	4.72	1.54	Optimal	Confirm	Relapse BSI ESBL-producing *K. pneumoniae*	Failed
#8	60/M	4	Fournier gangrene	16.5	NSTI	*K. oxytoca*	8	PIP-TZB 13.5 g/day CI	6.87	1.83	Optimal	Decrease	Yes	Cured
#9	55/F	1	Hydronephrosis	40	BSI	ESBL-producing *E. coli*	8	PIP-TZB 18 g/day CI	23.30	6.31	Optimal	Decrease	Yes	Cured
#10	82/M	12	Bladder cancer	71	BSI	ESBL-producing *K. pneumoniae*	4	PIP-TZB 18 g/day CI	13.22	1.68	Optimal	Decrease	Yes	Cured

BSI: bloodstream infection; CCI: Charlson Comorbidity Index; CI: continuous infusion; CLCr: creatinine clearance; Css: steady-state concentrations; C_T_: threshold concentration; cUTI: complicated urinary tract infection; ECPA: expert clinical pharmacological advice; ESBL: extended-spectrum beta-lactamase; MIC: minimum inhibitory concentration; NA: not assessed; NSTI: necrotizing soft tissue infection; PIP-TZB: piperacillin–tazobactam; PK/PD: pharmacokinetic/pharmacodynamic; TDM: therapeutic drug monitoring. PK/PD target column: green box: optimal PK/PD target; red box: suboptimal PK/PD target. Microbiological eradication column: green box: microbiological eradication; red box: microbiological failure.

**Table 3 antibiotics-12-01388-t003:** Case-by-case demographic and clinical features of six urologic patients with documented Gram-negative infections treated with CI meropenem according to a TDM-guided ECPA program.

ID Case	Age/Sex	CCI	Hospital Admission Diagnosis	Average CLCr (mL/min/ 1.73 m^2^)	Type of Infection	Pathogen	MIC (mg/L)	Beta-Lactam Initial Dosing	*f*Css/MIC Ratio	PK/PD Target	ECPA Recommendation at First TDM	Microbiological Eradication	Clinical Outcome
#1	74/M	5	Hydronephrosis	46	cUTI + BSI	AmpC-producing *E. cloacae*	0.125	MER 500 mg q6h CI	111.48	Optimal	Confirm	Yes	Cured
#2	68/F	6	Bladder cancer	111.5	SSI	*B. fragilis*	0.125	MER 500 mg q6h CI	65.46	Optimal	Confirm	Yes	Cured
#3	69/M	8	Bladder cancer	110.5	cUTI	*E. coli*	0.125	MER 500 mg q6h CI	35.12	Optimal	Increase	Yes	Cured
#4	72/F	9	Bladder cancer	21	cUTI	AmpC-producing *S. marcescens*	1	MER 500 mg q6h CI	26.17	Optimal	Decrease	Relapse BSI KPC-producing *S. marcescens*	Failed
#5	73/M	7	Bladder cancer	97	cUTI	*P. aeruginosa*/ESBL-producing *E. coli*	0.125	MER 1000 mg q6h CI	370.77	Optimal	Decrease	Yes	Cured
#6	75/F	5	Bladder cancer	84	BSI	ESBL-producing *K. pneumoniae*	0.125	MER 500 mg q6h CI	149.04	Optimal	Decrease	Yes	Cured

BSI: bloodstream infection; CCI: Charlson Comorbidity Index; CI: continuous infusion; CLCr: creatinine clearance; Css: steady-state concentrations; cUTI: complicated urinary tract infection; ECPA: expert clinical pharmacological advice; ESBL: extended-spectrum beta-lactamase; MER: meropenem; MIC: minimum inhibitory concentration; PK/PD: pharmacokinetic/pharmacodynamic; SSI: surgical site infection; TDM: therapeutic drug monitoring. PK/PD target column: green box: optimal PK/PD target; Microbiological eradication column: green box: microbiological eradication; red box: microbiological failure.

## Data Availability

The data presented in this study are available on request from the corresponding author. The data are not publicly available due to privacy concerns.

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
