# Peer review of "Pharmacokinetic/Pharmacodynamic Target Attainment of Continuous Infusion Piperacillin–Tazobactam or Meropenem and Microbiological Outcome among Urologic Patients with Documented Gram-Negative Infections"

_antibiotics, 2023, doi:10.3390/antibiotics12091388_

Round 1

Reviewer 1 Report

Pharmacokinetic/pharmacodynamic target attainment of continuous infusion piperacillin-tazobactam or meropenem and microbiological outcome among urologic patients with documented Gram-negative infections.

Thank you, this is an interesting study about the use of TDM guided treatment with piperacillin-tazobactam and meropenem. The study is retrospective and in a small number of patients and so has some limitations, but it does show that TDM may be useful in guiding therapy.

I have a few questions for clarity.

1.       Regarding the patients. Please could you provide more information about the patients? The importance of this lies in the changes to PK of all drugs caused by critical illness.

a.       Where were they being looked after, on a general/urology ward or a high dependency/intensive care unit?

b.       A Charlson co-morbidity score is provided but this only tells us about the patient before the current problem. The value of 5 suggests the score was made up of age plus one or two co-morbidities for most patients, but tells us little about their acute condition.

c.       How sick were the patients from their infections?  An APACHE II score (usually done in the first 24 hours of ICU admission) would help describe severity of illness, alternatively the sequential organ failure assessment (SOFA) score may be helpful.

d.       Did the patients have sepsis, Some, indication of white cell count/CRP/PCT would be helpful.

e.       How many of the patients were requiring support in intensive care?

f.        Was their renal dysfunction acute or chronic? This is important because the creatinine clearance calculations have little meaning if a patient has an AKI but are relevant of you are describing their chronic renal function? If you are describing AKI, please be clear about this and use the KDIGO criteria to describe severity please.

g.       How many patients developed an AKI during their admission (based on KDIGO criteria)?

h.       What were the albumin concentrations for these patients at the time of the antibiotics?

2.       The antibiotics, targets and dosing are described well. For clarification

a.       In your graph you plot the Tazobactam concentrations in relation to the MIC, going for fCss/CT > 1. Is this correct? The purpose of the tazobactam is to mitigate the beta lactamase activity. Is there a MIC for this? You allude to this in your discussion but the text falls short of explaining how the tazobactam should be adjusted. Did you adjust any of the doses based on the tazobactam results? This is a very good point but it needs better explanation.

b.       You gave good doses of both meropenem and pip/taz as loading doses (currently believed to be good practice in order to reach MIC requirements quickly). The maintenance doses were then adjusted according to renal function.
From your description, the results described are mainly about the first sample taken for TDM at around 24 hours. Several patients required a decrease in dose. Is it possible that the timing of the TDM caught the effects of the loading dose with the risk that decreasing the maintenance dose would lead to low concentrations later?

c.       The number of patients is too small to describe outcomes in terms of biological cure and/or resistance patterns. A little more caution around this would be appropriate. 

Some minor word changing is needed. 

Author Response

RESPONSE TO REVIEWERS

Manuscript ID: antibiotics-2554116 entitled “Pharmacokinetic/pharmacodynamic target attainment of continuous infusion piperacillin-tazobactam or meropenem and microbiological outcome among urologic patients with documented Gram-negative infections” by Gatti et al.

Dear Editor,

We would like to thank you for the opportunity to resubmit a revised version of this manuscript. We appreciated the reviewers’ constructive comments. All have been carefully considered and incorporated, where and whenever possible, in the revision.

Our point-by-point responses are provided below.

Q= QUERY; A= ANSWER

Reviewer #1

Thank you, this is an interesting study about the use of TDM guided treatment with piperacillin-tazobactam and meropenem. The study is retrospective and in a small number of patients and so has some limitations, but it does show that TDM may be useful in guiding therapy.

We thank the reviewer for appreciating our manuscript.

I have a few questions for clarity.

Q1.       Regarding the patients. Please could you provide more information about the patients? The importance of this lies in the changes to PK of all drugs caused by critical illness.

A1. Thank you for this comment. We provided more detailed information about our cohort according to your specific questions.

Q2.    Where were they being looked after, on a general/urology ward or a high dependency/intensive care unit?

A2. We thank the reviewer for this comment, allowing us to better clarify this issue. All patients were admitted to urology ward and then managed in the same ward. None of these required high dependency or ICU admission. We better clarified this issue in Methods section (refer to Line 194-195).

Q3.    A Charlson co-morbidity score is provided but this only tells us about the patient before the current problem. The value of 5 suggests the score was made up of age plus one or two co-morbidities for most patients, but tells us little about their acute condition.

A3. We thank the reviewer for this comment, and we agree with it. We added in Table 2 and Table 3 a specific column concerning reasons for hospital admission. Furthermore, we reported them also in the Results section (refer to Line 107-108).

Q4.    How sick were the patients from their infections?  An APACHE II score (usually done in the first 24 hours of ICU admission) would help describe severity of illness, alternatively the sequential organ failure assessment (SOFA) score may be helpful.

A4. We thank the reviewer for this comment, allowing us to better clarify this point. Considering that none of the patients required ICU admission and all were managed in urology ward, the calculation of APACHE II score or SOFA score at infection onset was not applicable.

Q5.    Did the patients have sepsis, Some, indication of white cell count/CRP/PCT would be helpful.

A5. Thank you for this suggestion. We added in Table 1 the median value of white cell count, CRP, and PCT as suggested.

Q6.   How many of the patients were requiring support in intensive care?

A6. Thank you for this comment. As previously specified in response to comment no. 2 and 4, all patients were managed during overall hospitalization in urology ward and none of them required ICU admission. We better specified this issue in Methods section (refer to Line 194-195).

Q7. Was their renal dysfunction acute or chronic? This is important because the creatinine clearance calculations have little meaning if a patient has an AKI but are relevant of you are describing their chronic renal function? If you are describing AKI, please be clear about this and use the KDIGO criteria to describe severity please.

A7. We thank the reviewer for this important comment, and we agree with the fact that creatinine clearance calculations have negligible meaning in patients developing sepsis-associated AKI, conversely playing a major role in describing patients with chronic renal function. Indeed, our cohort consisted of patients with chronic renal failure, considering that more than half of patients had a baseline CLCr below 50 mL/min/1.73m2, being the median CLCr 41 mL/min/1.73m2. None of the included patients developed sepsis-associated AKI, thus KDIGO criteria were not applicable. We better specified this issue in Results section (refer to Line 106-107).

Q8.  How many patients developed an AKI during their admission (based on KDIGO criteria)?

A8. We thank the reviewer for this comment. As specified in comment no. 7, none of the patients developed sepsis-related AKI, thus KDIGO criteria were not applicable. Reported CLCr calculations referred to chronic renal failure affecting more than half of included patients, as better specified in Results section (refer to Line 104-107).

Q9. What were the albumin concentrations for these patients at the time of the antibiotics?

A9. Thank you for this suggestion. We added in Table 1 the median value of albumin during antibiotic course as suggested. However, it is important to highlight that, according to the low-moderate plasmaprotein binding for both piperacillin-tazobactam and meropenem (20-23% and 2%, respectively, as reported in the Methods section), the impact of hypoalbuminemia on free antibiotic concentrations is negligible compared to agents with high (>90%) protein binding. We added a specific comment in the Discussion section (refer to Line 185-187).

Q10.    The antibiotics, targets and dosing are described well. For clarification

In your graph you plot the Tazobactam concentrations in relation to the MIC, going for fCss/CT > 1. Is this correct? The purpose of the tazobactam is to mitigate the beta lactamase activity. Is there a MIC for this? You allude to this in your discussion but the text falls short of explaining how the tazobactam should be adjusted. Did you adjust any of the doses based on the tazobactam results? This is a very good point but it needs better explanation.

A10. We thank the reviewer for this relevant comment, allowing us to better clarify this important issue. As also reported in response to comment no. 4 of reviewer 2, preclinical evidence identified the f%T>CT as the PK/PD parameter most closely associated with bacterial kill with tazobactam (refer to doi: 10.1002/phar.2210). The threshold concentration was fixed to 4 mg/L according to the value adopted by the EUCAST for tazobactam when testing piperacillin-tazobactam susceptibility. CT doesn’t correspond to a real MIC, but represents the concentration by which beta-lactamase inhibitory activity is maximized. As reported in Methods section (refer to Line 278-282), fail in attaining a tazobactam fCss/CT >1 leads to the achievement of quasi-optimal or suboptimal according to joint PK/PD target concept. Consequently, we adjusted piperacillin-tazobactam dosing not only on the basis of piperacillin concentrations, but also tazobactam concentrations were taken into account, being dosing increases required whenever tazobactam concentrations are below 4 mg/L (i.e., fCss/CT < 1). In regard to our findings, it is noteworthy that CI administration may be a useful tool for maintaining concentrations steadily over time above the safeguarded fixed threshold of 4 mg/L. We clarified this issue in Methods (refer to Line 273-275) and Discussion sections (refer to Line 179-181).

Q11.  You gave good doses of both meropenem and pip/taz as loading doses (currently believed to be good practice in order to reach MIC requirements quickly). The maintenance doses were then adjusted according to renal function.

From your description, the results described are mainly about the first sample taken for TDM at around 24 hours. Several patients required a decrease in dose. Is it possible that the timing of the TDM caught the effects of the loading dose with the risk that decreasing the maintenance dose would lead to low concentrations later?

A11. We thank the reviewer for this interesting comment, allowing us to better clarify this important point. Considering the short half-life of both piperacillin-tazobactam and meropenem (approximatively 1-1.5 hours), it is unlikely that the need for beta-lactam dosing decreases was related to the administered loading doses also for patients with severe renal dysfunction. Furthermore, it is noteworthy that none of the included patients required dosing increase at TDM reassessment (refer to Table 1), thus supporting the fact that decreasing of maintenance at first TDM-guided ECPA didn’t lead to lower beta-lactam concentrations.

Q12.     The number of patients is too small to describe outcomes in terms of biological cure and/or resistance patterns. A little more caution around this would be appropriate.

A12. We thank the reviewer for this comment, and we agree with the fact that considering the limited sample size, a little more caution would be appropriate in describing the relationship between PK/PD target attainment and microbiological outcome and/or resistance development. We modified accordingly the Discussion section (refer to Line 151-152 and Line 319).

Reviewer 2 Report

The authors have conducted an interesting study; the topic is of immense importance in the current context of AMR and gives useful insights into TDM based approach for dose optimisation of antibiotics. Few points worth mentioning are as under:

1.     Introduction: Please cite some studies reporting TDM based approach for the beta-lactams or other antibiotics in similar or other clinical scenarios?

2.     The prescription of piperacillin/tazobactam or meropenem to patients was at the discretion of treating physician. What were the criteria for selection of a particular agent?

3.     The initial MD regimen was defined according to renal function and underlying pathophysiological condi tions of each included patient”: Details of the protocols for defining the initial maintenance dosage (MD) of both drugs may be included.

4.     What was the defined “threshold concentration” for tazobactam? The authors mention that they introduced the concept of joint PK/PD target attainment by measuring concentration of tazobactam in addition to piperacillin. This needs to be discussed elaboratively especially in the context of their findings.

5.     The concept of extended or continuous infusion of beta-lactams has mainly emerged due to better outcomes compared to short term intravenous infusion regimen. In the study, all the patients were given continuous infusion (CI) of both the agents. Was this a routine protocol or extended infusion protocol is also followed depending on case-to-case basis? What is the evidence on efficacy and safety of extended versus continuous infusion of beta-lactams? The authors may add relevant text with supporting evidence under discussion.

6.     If possible (due to retrospective design), addition of data on clinical outcomes will enhance the quality of results in addition to the microbiological outcomes already mentioned. If data on clinical outcomes is not available, the same may be mentioned  under study limitations.  

Minor editing required.

Author Response

RESPONSE TO REVIEWERS

Manuscript ID: antibiotics-2554116 entitled “Pharmacokinetic/pharmacodynamic target attainment of continuous infusion piperacillin-tazobactam or meropenem and microbiological outcome among urologic patients with documented Gram-negative infections” by Gatti et al.

Dear Editor,

We would like to thank you for the opportunity to resubmit a revised version of this manuscript. We appreciated the reviewers’ constructive comments. All have been carefully considered and incorporated, where and whenever possible, in the revision.

Our point-by-point responses are provided below.

Q= QUERY; A= ANSWER

Reviewer #2

The authors have conducted an interesting study; the topic is of immense importance in the current context of AMR and gives useful insights into TDM based approach for dose optimisation of antibiotics. Few points worth mentioning are as under:

We thank the reviewer for appreciating our study.

Q1.     Introduction: Please cite some studies reporting TDM based approach for the beta-lactams or other antibiotics in similar or other clinical scenarios?

A1. We thank the reviewer for this relevant suggestion. We added a specific statement highlighting the advantages of optimizing beta-lactams by means of a TDM-guided approach compared to standard management in improving clinical outcome in different clinical scenarios (refer to Line 63-66 and references 13-17).

Q2.     The prescription of piperacillin/tazobactam or meropenem to patients was at the discretion of treating physician. What were the criteria for selection of a particular agent?

A2. Thank you for this comment. We better clarified criteria for selection of the specific agents in Methods section as suggested (refer to Line 229-234).

Q3.    “The initial MD regimen was defined according to renal function and underlying pathophysiological conditions of each included patient”: Details of the protocols for defining the initial maintenance dosage (MD) of both drugs may be included.

A3. We thank the reviewer for this comment. We added in Methods section (refer to Line 239-245) details concerning initial maintenance dose of both piperacillin-tazobactam and meropenem according to renal function.

Q4.   What was the defined “threshold concentration” for tazobactam? The authors mention that they introduced the concept of joint PK/PD target attainment by measuring concentration of tazobactam in addition to piperacillin. This needs to be discussed elaboratively especially in the context of their findings.

A4. We thank the reviewer for this relevant comment, allowing us to better clarify this important issue. Preclinical evidence identified that  f%T>CT was the PK/PD parameter most closely associated with bacterial kill of tazobactam (refer to doi: 10.1002/phar.2210). The threshold concentration of 4 mg/L was selected for tazobactam according to the value adopted by the EUCAST when testing piperacillin-tazobactam susceptibility. In regard to our findings, it is noteworthy that CI administration may be a useful tool for maintaining concentrations steadily over time above the safeguarded fixed threshold of 4 mg/L. We clarified this issue in Methods (refer to Line 273-275) and Discussion sections (refer to Line 179-181).

Q5.   The concept of extended or continuous infusion of beta-lactams has mainly emerged due to better outcomes compared to short term intravenous infusion regimen. In the study, all the patients were given continuous infusion (CI) of both the agents. Was this a routine protocol or extended infusion protocol is also followed depending on case-to-case basis? What is the evidence on efficacy and safety of extended versus continuous infusion of beta-lactams? The authors may add relevant text with supporting evidence under discussion.

A5. We thank the reviewer for this comment, allowing us to better clarify this important issue and further emphasize the relevant role of continuous infusion in improving clinical outcome compared to intermittent infusion. At our institution, beta-lactams are routinely administered by continuous infusion in all of the different clinical scenarios (i.e., ICU, hematology, pediatrics, medical and surgical wards). According to time-dependency, the need for promptly attaining aggressive PK/PD targets for minimizing resistance development, and the several evidence reporting a significant lower mortality rate and higher clinical/microbiological cure rate compared to intermittent infusion, as reported in the already mentioned references 19-23. We further detailed findings retrieved in these studies supporting the advantages of extended/continuous infusion of beta-lactams in Discussion section (refer to Line 161-163).

Q6.     If possible (due to retrospective design), addition of data on clinical outcomes will enhance the quality of results in addition to the microbiological outcomes already mentioned. If data on clinical outcomes is not available, the same may be mentioned  under study limitations.

A6. We thank the reviewer for this comment. We added data on clinical outcome in Table 1-3 and in the Results section (refer to Line 140-141). The specific definition of clinical cure was added in the Methods section (refer to Line 304-307).